# Associations of resuscitation fluid load, colloid-to-crystalloid infusion ratio and clinical outcomes in children with dengue shock syndrome

Vo Thanh Luan[1☯], Vo Thi-Hong Tien[1], Ngo Thi-Mai Phuong[1,2], Do Chau Viet[1], Trinh Huu Tung[1,3], Nguyen Tat Thanh[1,4☯]*

1 Department of Infectious Diseases, Children's Hospital 2, Ho Chi Minh City, Vietnam, 2 Department of Family Medicine, University of Medicine and Pharmacy, Ho Chi Minh City, Vietnam, 3 Department of Pediatric Cardiology, Children's Hospital 2, Ho Chi Minh City, Vietnam, 4 Department of Tuberculosis, Woolcock Institute of Medical Research, Ho Chi Minh City, Vietnam

☯ These authors contributed equally to this work.
* thanhhonor@gmail.com, thanh.tat.nguyen@sydney.edu.au

**Data Availability Statement:** The authors confirm that all data underlying the findings are fully available without restriction. All relevant data are

## Abstract

### Background

Severe respiratory distress and acute kidney injury (AKI) are key factors leading to poor outcomes in patients with dengue shock syndrome (DSS). There is still limited data on how much resuscitated fluid and the specific ratios of intravenous fluid types contribute to the development of severe respiratory distress necessitating mechanical ventilation (MV) and AKI in children with DSS.

### Methodology/principal findings

This retrospective study was conducted at a tertiary pediatric hospital in Vietnam between 2013 and 2022. The primary outcomes were the need for MV and renal function within 48 h post-admission. A predictive model for MV was developed based on covariates from the first 24 h of PICU admission. Changes in renal function within 48 h were analyzed using a linear mixed-effects model. A total of 1,278 DSS children with complete clinical and fluid data were included. The predictive performance of MV based on the total intravenous fluid volume administered yielded an AUC of 0.871 (95% CI, 0.836–0.905), while the colloid-to-crystalloid ratio showed an AUC of 0.781 (95% CI, 0.743–0.819) (both $P < 0.001$). The optimal cut-off point of the cumulative fluid infusion was 181 mL/kg, whereas that of the colloid-to-crystalloid ratio was 1.6. Multivariable analysis identified female patients, severe bleeding, severe transaminitis, excessive fluid resuscitation, and a higher proportion of colloid solutions in the first 24 h as significant predictors of MV in DSS patients. The predictive model for MV demonstrated high accuracy, with a C-statistic of 89%, strong calibration, and low Brier score (0.04). Importantly, a more pronounced decline in glomerular filtration rate was observed in DSS patients who required MV than in those who did not.

within the paper and its Supporting Information files.

**Funding:** The author(s) received no specific funding for this work.

**Competing interests:** The authors have declared that no competing interests exist.

## Conclusions/significance

This study provides insights into optimizing fluid management protocols, highlighting the importance of monitoring fluid volume and the colloid-to-crystalloid ratio during early resuscitation to improve the clinical outcomes of DSS patients.

## Author summary

Dengue shock syndrome (DSS) in children admitted to the pediatric intensive care unit (PICU) has mortality rates ranging from 5% to > 20%, particularly in those with prolonged shock, severe respiratory distress requiring mechanical ventilation (MV), and acute kidney injury (AKI). Although colloidal solutions are commonly used in severe DSS management, excessive use can lead to respiratory failure and AKI, raising concerns regarding optimal fluid strategies. This study analyzed 1,278 children with DSS and found that 13.3% required MV, with a significantly higher mortality rate in the MV group (22.4%) than in the non-MV group (0.1%). Both the total intravenous fluid volume and colloid-to-crystalloid ratio were strong predictors of MV, with optimal cut-offs of 181 mL/kg for fluid volume and a ratio of 1.6. In practice, a cut-off of colloid-to-crystalloid ratio $\geq$ 1.6 may indicate the need to transition to alternative fluids such as albumin or fresh frozen plasma. Moreover, a marked reduction in the estimated glomerular filtration rate (eGFR) was observed in the MV group, indicating a link between fluid management and renal function. These findings underscore the importance of fluid management in DSS, providing key predictors of MV and AKI to guide treatment and improve the clinical outcomes.

## Introduction

Dengue infection spans a spectrum from asymptomatic cases to fatalities, and dengue shock syndrome (DSS) stands out as the most common life-threatening complication [1]. Patients with DSS admitted to the pediatric intensive care unit (PICU) have varied mortality rates, ranging from 5% to 20% [1,2]. Mortality rates increase to > 30% in patients experiencing prolonged DSS, severe bleeding, acute liver failure, acute kidney injury (AKI), and severe respiratory failure requiring mechanical ventilation (MV) [3–6]. Notably, a cumulative intravenous fluid overload exceeding 10–15% is highly associated with respiratory failure and mortality in children with severe dengue cases [3–6]. This highlights the critical need for precise fluid resuscitation to reduce mortality in children with DSS. The World Health Organization (WHO) 2009 dengue guidelines have recommended minimal fluid administration to ensure adequate tissue perfusion and urine output > 0.5 ml/kg/h as an indicator for tapering fluid infusion [7]. Colloidal solutions are often used to manage patients with severe and prolonged DSS, manifesting as progressive plasma leakage and a poor response to crystalloids [8]. Likewise, the early requirement for colloids over crystalloids indicates a significant plasma loss or prolonged dengue shock [7,8]. However, colloids are recommended for prudent use because of their numerous adverse effects [7]. In addition, excessive colloid infusion can lead to AKI owing to lysosomal accumulation in the proximal tubules, resulting in tubular swelling and reduced renal blood flow pressure [9,10]. Furthermore, other critical factors such as severe bleeding, liver injury, and profound DSS can contribute to AKI risk but are not well documented. In particular, the effects of colloids on the dynamic variation of glomerular filtration rate (GFR) in patients with DSS remain limited in the current literature [11–14]. Therefore, an

excessive intravenous fluid volume of either crystalloids or colloids is associated with poor clinical outcomes [3–6]. However, to date the optimal volume of resuscitated fluid and the appropriate colloid-to-crystalloid ratio for managing patients with DSS remain unclear.

Previous studies have shown that severe respiratory failure requiring MV is a strong independent predictor of death in hospitalized children with severe dengue [2,3,6]. We have recently reported that cumulative infused fluid from referral hospitals and 24 hours of PICU admission was a significant risk factor for MV support [2]. Therefore, we hypothesized that a larger volume of resuscitation fluid and more colloids than infused crystalloids would lead to poor clinical outcomes, including mechanical ventilation requirement and acute kidney injury among hospitalized children with DSS. This study aimed to investigate the associations between resuscitation fluid load, the colloid-to-crystalloid infusion ratio, and clinical outcomes in children with DSS admitted to the PICU. A comprehensive understanding of fluid management strategies, particularly in patients with DSS requiring MV, is essential to improve survival outcomes. We aimed to establish an optimized fluid management protocol to minimize the need for MV and reduce the risk of kidney injury in pediatric DSS patients.

## Methods

### Ethics statement

This sub-study stems from the main retrospective research titled "Prognostic model predicting mortality among patients presenting with dengue shock syndrome (DSS) at Children's Hospital 2, during 2013–2022," approved by the Scientific Committee and Institutional Review Board (IRB) of Children's Hospital 2, Ho Chi Minh City, Vietnam (IRB No. 893/QD-BVND2, signed on June 6, 2022) [15]. Using a secondary dataset from this primary research posed less than minimal risk to the participants [15]. Therefore, the requirement for informed consent was waived, and patient identities were anonymized to ensure confidentiality in accordance with the ethical guidelines of the Declaration of Helsinki.

### Study setting and population

The primary study was conducted at the Children's Hospital 2, a tertiary pediatric hospital in southern Vietnam, spanning from 2013 to 2022 [15]. The eligibility criteria included children under 18 years of age with laboratory-confirmed dengue infection and clinical presentation of DSS, following the WHO 2009 dengue guidelines [7]. Exclusion criteria were the absence of serological confirmation of dengue infection and incomplete fluid resuscitation data.

### Study definitions

Dengue infection confirmation was based on the WHO 2009 criteria, relying on positive results for the non-structural 1 (NS1) antigen or Dengue-IgM antibody test [7]. Severe transaminitis was defined as elevated aspartate aminotransferase (AST) and/or alanine aminotransferase (ALT) levels $\geq$ 1,000 IU/L [7]. The definition of profound DSS involved two criteria: (i) the requirement of more than two colloid boluses during the first episode of compensated DSS or more than two episodes of recurrent shock or (ii) the need for colloid resuscitation combined with inotropes to maintain stable hemodynamics [4,16]. Mechanical ventilation indications adhered to the Vietnamese Ministry of Health Dengue Guidelines, as outlined in **S1 File**.

### Study outcomes and candidate variables

The main study outcomes were mechanical ventilation requirement during PICU stay and changes in renal function within 48 h after PICU admission. A predefined set of covariates was

predetermined, including age, sex, DSS severity, severe bleeding, severe transaminitis, platelet counts, hematocrit levels, cumulative infused fluid volume within the first 24 h of PICU admission, and the percentages of colloid and crystalloid fluids. These covariates were preselected based on our clinical knowledge, disease pathogenesis and medical literature [2,4,15]. Notably, obesity data were missing in approximately 40% of participants with available intravenous fluid infusion data; therefore, the obesity covariate was excluded from the analysis.

## Data collection and measurements

Clinical and laboratory data were collected at various time points from paper-based hospital medical records and entered into structured case-report forms. The dataset in **S1 Apppendix** included comprehensive information on age, sex, DSS severity, severe bleeding, liver transaminase levels, platelet counts, hematocrit levels, and the cumulative volume of infused fluids during the first 24 hours of PICU admission. Systolic and diastolic shock indices were used to assess dengue severity and the need for vasoactive inotrope support, as indicated by higher values associated with more severe disease and vasopressor requirements [4,16]. Blood and chemical tests including serum creatinine levels, were performed using an Alinity-ci-series machine (Abbott, USA). The estimated glomerular filtration rate (eGFR) was calculated using the Schwartz equation: estimated GFR = 36.5 × height (cm) / serum creatinine (μmol/L) [17]. Only the participants with complete datasets were included in the final analysis. This study is reported in accordance with the guidelines of Strengthening the Reporting of Observational studies in Epidemiology (**S1 Checklist**).

## Statistical analysis

Continuous variables were described using median and interquartile ranges (IQRs), while categorical variables were presented as numerical counts and percentages (%). The primary study bias was missing data on serum creatinine and patient height for eGFR estimation from retrospective data collection. To address this issue, only the participants with complete data were included in the final analysis. The receiver operating characteristic (ROC) curve with area under the curve (AUC) values, sensitivity, specificity, and accuracy was evaluated to predict the risk of mechanical ventilation and changes in renal function within 48 h post-PICU admission. Univariable and multivariable logistic regression analyses were performed using complete-case analysis based on predefined sets of covariates chosen according to disease pathogenesis and clinical experience. The cumulative fluid volume from referral hospitals and within the first 24 h of PICU admission was log2-transformed for standardization. Additionally, the colloid-to-crystalloid infusion ratio was standardized using a square root transformation. Backward stepwise model selection, guided by the Akaike Information Criteria (AIC), was applied to determine the best predictive model, including interactions among covariates. Internal validation was performed using bootstrap resampling (n = 500). Model performance was assessed using the C-statistic, calibration slope, and Brier score [18,19]. Furthermore, a linear mixed-effects model was employed to explore the associations between the total intravenous fluid load, colloid-to-crystalloid ratio, and renal function changes within 72 h post-admission. All statistical tests were two-sided, with statistical significance set at $P < 0.05$. Data analyses were conducted using R statistical software (version 4.3.2, Boston, MA, USA).

## Results

### Baseline characteristics of study participants upon PICU admission

Between 2013 and 2022, approximately 2,000 children with DSS were admitted to the PICU, of whom 1,278 met the eligibility criteria and were included in the analysis (**Fig 1**). The clinical

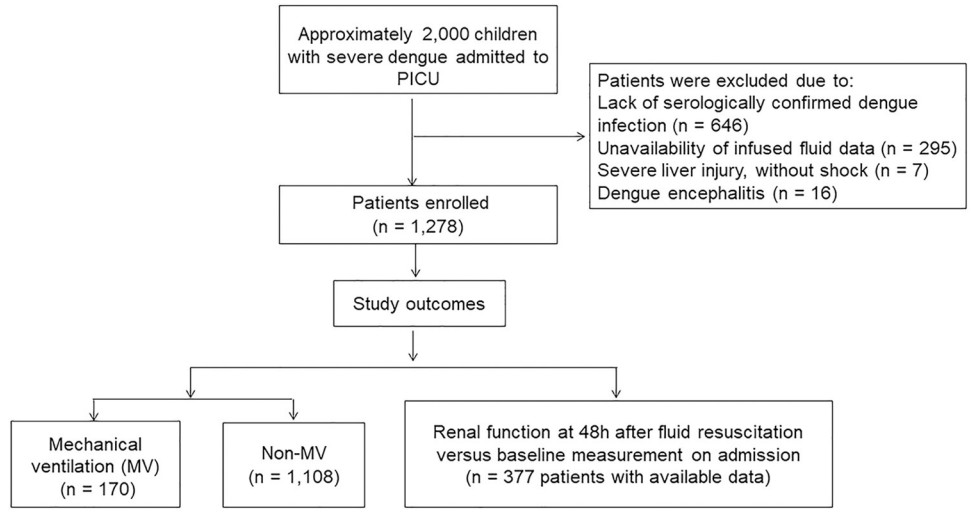

**Fig 1. The flowchart of study participants.**

and laboratory data of the participants upon PICU admission are presented in **Table 1**. The median patient age was 8.1 years (interquartile range, IQR: 5.4–10.7), and females accounted for 48% of all patients. The median body mass index was 18.1 (IQR, 15.6–21.4) kg/m$^2$, and 80 (6%) patients had underlying diseases. Participants experienced DSS on the median day 5 (IQR: 4–5 days) after the onset of fever. Notably, 1,171 (92%) children were diagnosed with compensated DSS and the remaining 107 (8%) patients had decompensated DSS. The systolic and diastolic shock indices were markedly elevated. Critical bleeding was observed in 95 (7.4%) patients. Complete blood counts showed a marked increase in hematocrit and decrease in platelet cell count. Severe transaminitis was observed in 130 (10.2%) participants. The median serum lactate was 2.3 (IQR, 1.7–3.3) mmol/L. Most notably, PICU-admitted patients with DSS received a high cumulative amount of intravenous (IV) fluids (crystalloid and colloid solutions) infused from referral hospitals and 24h PICU admission with a median of 135 (IQR, 105–181) mL/kg, and median ratio of colloid to crystalloid solution infused was 0.72 (IQR, 0–2.1). Additionally, 97 (7.6%) need vasopressors during the first 24h of admission, with a median vasoactive inotropic score (VIS) was 20 (IQR, 10–42.5).

## Clinical outcomes of study participants

On admission, patients exhibited a slight increase in respiratory rate (median 25 breaths/min; IQR, 22–30) and mild abnormalities in blood gas analysis. Despite this, 170 (13.3%) patients progressed to severe respiratory failure, requiring mechanical ventilation within 48 hours of PICU admission (**Table 1**). The estimated glomerular filtration rate (eGFR) showed a slight improvement at 48 h post-admission (median 94 mL/min; IQR, 76–108 mL/min) compared to admission values (median 86 mL/min; IQR, 72–100 mL/min).

Overall, 39 of 1,278 (3.1%) DSS patients died during the PICU stay, with a median hospital stay of 4 days (IQR, 3–6). Among those requiring mechanical ventilation, the mortality rate was significantly higher (38 deaths among 170 MV patients, 22.4%). The primary causes of death included acute respiratory distress syndrome (ARDS), pulmonary and cerebral hemorrhage, dengue-associated acute liver failure, multiorgan failure, and severe PICU-acquired infections.

**Table 1. Clinical and laboratory characteristics of study participants on PICU admission and associated clinical outcomes within 48h post-admission and at discharge (N = 1,278).**

| Characteristics | Summary statistics |
|---|---|
| ***On PICU admission*** | |
| Age (years) | 8.1 (5.4–10.7) |
| Sex: Female (%) | 619 (48) |
| Body mass index (kg/m$^2$) | 18.1 (15.6–21.4) |
| Underlying diseases (%) | 80 (6) |
| Day of DSS since disease onset (days) | 5 (4–5) |
| Grading of DSS severity (%)<br>Compensated DSS<br>Decompensated DSS | <br>1,171 (92)<br>107 (8) |
| Severe bleeding (%) | 95 (7.4) |
| Respiratory rate (/min) | 25 (22–30) |
| Systolic shock index (bpm/mmHg) | 1.30 (1.11–1.5) |
| Diastolic shock index (bpm/mmHg) | 1.69 (1.44–2.0) |
| White blood cell count (x 10$^9$/L) | 4.8 (3.37–6.8) |
| Hemoglobin (g/dL) | 14.9 (13.4–16.1) |
| Peak hematocrit (%) | 48 (45–51) |
| Nadir hematocrit (%) | 38 (35–41) |
| Platelet counts (x 10$^9$/L) | 36 (23–55) |
| Aspartate aminotransferase (IU/L) | 151 (87–346) |
| Alanine aminotransferase (IU/L) | 66 (35–169) |
| Severe transaminitis (%) | 130 (10.2) |
| International normalized ratio (INR) | 1.23 (1.1–1.49) |
| Troponin I (ng/mL) | 0.012 (0.01–0.054) |
| Serum lactate (mmol/L) | 2.3 (1.7–3.3) |
| Arterial blood gas analysis<br>pH<br>PCO$_2$ (mmHg)<br>PO$_2$ (mmHg)<br>Bicarbonate (mEq/L) | <br>7.44 (7.40–7.48)<br>26.1 (22–30.2)<br>111 (65–149)<br>17.2 (14.7–19.7) |
| Cumulative fluid infused from referral hospitals and 24h PICU admission (mL/kg) | 135 (105–181) |
| Ratio of colloid-to-crystalloid infusion | 0.72 (0–2.1) |
| Vasopressor support during first 24h (%) | 97 (7.6) |
| Vasoactive inotropic score | 20 (10–42.5) |
| **Clinical outcomes of patients** | |
| ***Within first 48h of PICU admission*** | |
| Mechanical ventilation support (%) | 170 (13.3) |
| eGFR on admission (mL/min) | 86 (72–100) |
| eGFR at 48h post-admission (mL/min) | 94 (76–108) |
| ***At hospital discharge*** | |
| Length of hospital stay (days) | 4 (3–6) |
| Fatal outcome (%) | 39 (3.1) |

Summary statistics are presented as median (interquartile range, IQR) for continuous variables and frequency (%) for categorical variables. **Abbreviations:** DSS, dengue shock syndrome; eGFR, estimated glomerular filtration rate by Schwartz equation; PICU, pediatric intensive care unit

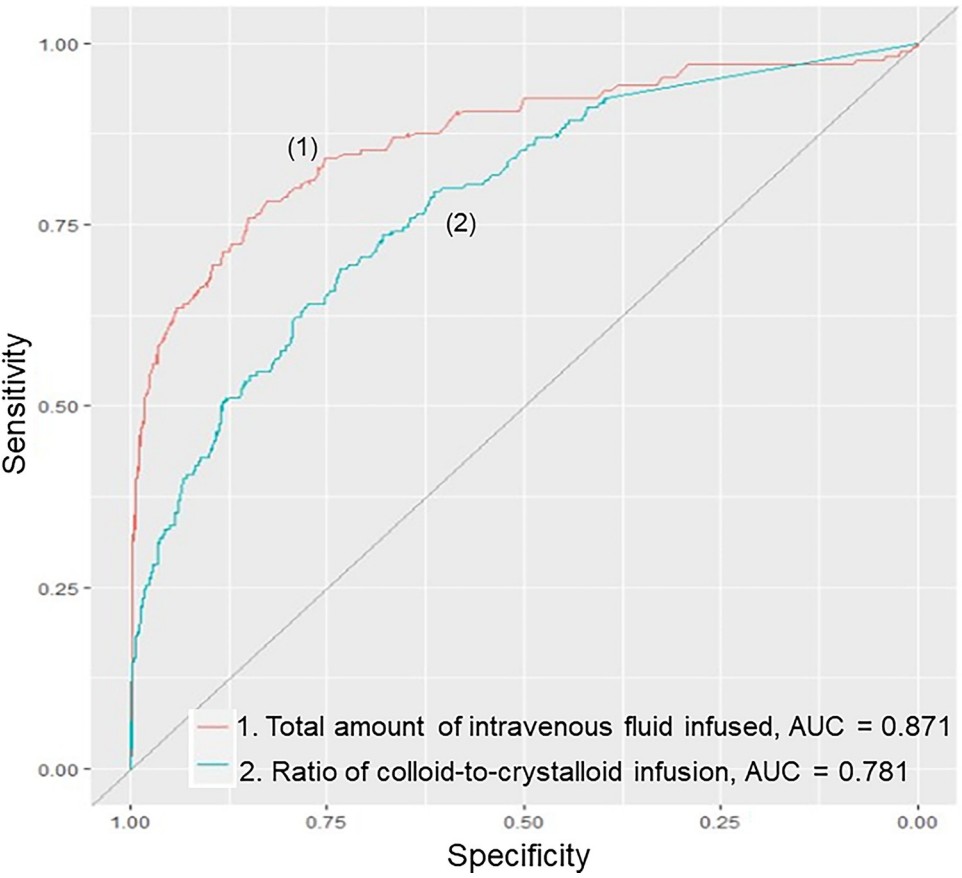

**Fig 2. Area under the receiver operating characteristic curves (AUCs) for the cumulative amount of intravenous fluid infusion and colloid-to-crystalloid ratio in predicting the need for MV.**

## Predicting the risk of mechanical ventilation, based on fluid resuscitation data during the first 24 h of admission

The performance of prognostic indicators for predicting mechanical ventilation (MV) in DSS children is summarized in **Fig 2** and **Table 2**. A large volume of intravenous fluids infused

**Table 2. Area under the curve for cumulative infused fluid from referral hospitals and during the first 24h of admission and cutoffs for mechanical ventilation.**

| Parameters | AUC of MV support (95% confidence interval) | | *P*-value |
|---|---|---|---|
| Cumulative IV fluid infused from referral hospitals and 24h PICU admission (mL/kg) | 0.871 | 0.836–0.905 | < 0.001 |
| Colloid fluid (%) | 0.701 | 0.664–0.739 | < 0.001 |
| Crystalloid fluid (%) | 0.842 | 0.811–0.872 | < 0.001 |
| Blood products (%) [a] | 0.900 | 0.870–0.930 | < 0.001 |
| Ratio of colloid-to-crystalloid infusion | 0.781 | 0.743–0.819 | < 0.001 |
| **Optimal cutoff points** | **Sensitivity** | **Specificity** | **Accuracy** |
| Total volume of fluid infusion ≥ 181 (mL/kg) | 0.782 | 0.828 | 0.822 |
| Ratio of colloid-to-crystalloid infusion ≥ 1.6 | 0.688 | 0.735 | 0.729 |

**Abbreviation:** AUC, Area under the curve; MV, Mechanical ventilation; PICU, Pediatric intensive care unit

[a] Blood products included packed red blood cells, platelet transfusion, fresh frozen plasma, and albumin solutions

from referral hospitals and during the first 24 hours of PICU admission was a strong predictor of MV, with an AUC of 0.871 (95% CI, 0.836–0.905, $P < 0.001$). A higher colloid-to-crystalloid infusion ratio was also a significant predictor with an AUC of 0.781 (95% CI, 0.743–0.819, $P < 0.001$). The blood products used in DSS patients with critical bleeding, low platelet counts, and severe plasma leakage demonstrated a high predictive value for MV. Further analysis identified the optimal cut-off points for these indicators. A total IV fluid infusion of $\geq$ 181 mL/kg had a sensitivity of 78.2% and specificity of 82.8% for predicting the requirement for MV support. The cut-off for the colloid-to-crystalloid infusion ratio was $\geq$ 1.6, with a sensitivity of 68.8% and a specificity of 73.5%. These findings underscore the importance of fluid management strategies in early DSS care to identify patients at risk for MV.

## Associations of resuscitation fluid load, colloid-to-crystalloid infusion ratio and the need for MV in children with DSS

As shown in **Table 3**, multivariable backward stepwise logistic regression analysis using AIC values identified key prognostic indicators for MV requirement in patients with DSS. Significant factors included female patients, severe bleeding, severe transaminitis, a large cumulative amount of fluid infusion from referral hospitals and within the first 24 h of PICU admission, and a high colloid-to-crystalloid infusion ratio. Interestingly, a high hematocrit level served as a protective factor. No significant interactions were found among the covariates.

## Performance of predictive model for mechanical ventilation and internal validation

The predictive model for mechanical ventilation in DSS patients exhibited strong performance (**Table 3**). It demonstrated high discrimination with a C-statistic of 89%, along with good calibration and low Brier scores in both training and test sets. The calibration plot (**Fig 3**) showed a high agreement between the predicted and observed outcomes, indicating the reliability of the model for predicting mechanical ventilation in this patient population.

## Changes in the estimated glomerular filtration rate by mechanical ventilation upon PICU admission and 48h post-admission

The dynamic variations in renal function at admission and 48 h post-admission in MV and non-MV patients are shown in **Fig 4A**. There was a significant difference in renal function between DSS patients requiring MV and those who did not. In the non-MV group, eGFR improved, whereas in the MV group, eGFR deteriorated. Further analysis revealed a significant decline in eGFR among MV patients from a mean of 81 mL/min on admission to 73 mL/min 48 h after PICU admission ($P = 0.01$, paired t-test). Conversely, DSS patients not requiring MV showed a significant improvement in eGFR, increasing from 92 mL/min on admission to 103 mL/min at 48 h after hospitalization ($P < 0.001$, paired t-test; **Fig 4B**). These findings suggest that DSS patients undergoing MV are more prone to kidney injury, as indicated by a reduced eGFR, than those who do not require MV.

## Associations of within 48h-admission dynamic changes in glomerular filtration rate, and resuscitation fluid load

As shown in **Table 4**, the linear mixed-effects model demonstrated dynamic changes in glomerular filtration rate (eGFR) from PICU admission to 48 hours post-admission in DSS patients. Random effect analysis revealed an intragroup correlation of 5%, indicating that the mechanical ventilation factor contributed to 5% of the variance in eGFR. The fixed-effects

**Table 3. Associations of resuscitation fluid load, colloid-to-crystalloid ratio and the need for mechanical ventilation in children with DSS.**

| Risk factors | MV (n = 170) | Non-MV (n = 1,108) | Unadjusted effect | Adjusted effect |
|---|---|---|---|---|
| | Summary statistic | Summary statistic | OR (95% CI), *P*-value | OR (95% CI), *P*-value |
| Age (years) | 6.9 (4–9) | 8.4 (5.7–10.9) | 0.89 (0.85–0.93), $P < .001$ | - |
| Patient sex (%) | | | | |
| Male | 75 (44) | 584 (53) | 1.42 (1.03–1.98), $P = .038$ | 1.88 (1.11–3.18), $P = .02$ |
| Female | 95 (56) | 524 (47) | | |
| Severe bleeding (%) [a] | | | | |
| No | 93 (55) | 1090 (98) | 56.7 (31.8–101), $P < .001$ | 20.74 (8.77–49.1), $P < .001$ |
| Yes | 77 (45) | 18 (2) | | |
| Severe transaminitis (%) [a] | | | | |
| No | 102 (60) | 1046 (94) | 11.1 (7.41–16.63), $P < .001$ | 6.21 (3.16–12.2), $P < .001$ |
| Yes | 68 (40) | 62 (6) | | |
| Hematocrit (%) | 47 (41–50) | 48 (45–51) | 0.91 (0.88–0.93), $P < .001$ | 0.89 (0.84–0.93), $P < .001$ |
| Platelet counts ($< 20 \times 10^9$/L), (%) | | | | |
| No | 120 (71) | 900 (82) | 1.87 (1.3–2.68), $P < .001$ | - |
| Yes | 50 (29) | 201 (18) | | |
| *Log2*-Cumulative infused fluid from referral hospitals and 24h PICU admission (mL/kg) [b] | 8.01 (7.55–8.48) | 6.99 (6.66–7.35) | 17.7 (11.78–26.62), $P < .001$ | 14.18 (8.1–24.96), $P < .001$ |
| *Square root*-Ratio of colloid-to-crystalloid infusion [c] | 1.76 (1.04–2.63) | 0.75 (0–1.31) | 3.45 (2.83–4.22), $P < .001$ | 2.2 (1.55–3.1), $P < .001$ |
| **Performance of multivariable model developed: internal validation [d]** | | | | |
| C-statistic | 0.89 (0.88) | | | |
| Calibration-in-the-large | 0 (- 0.03) | | | |
| Calibration slope | 1 (0.97) | | | |
| Brier score | 0.04 (0.04) | | | |

Summary statistics are median (IQR) for continuous variables and frequency (%) for categorical data. **Abbreviations**: CI, Confidence interval; DSS, Dengue shock syndrome; OR, Odds ratio, PICU, Pediatric intensive care unit.

[a] These factors were defined by the 2009 WHO dengue guidelines

[b] Standardized covariate of the cumulative amount of intravenous fluid from referral hospitals and within 24h PICU admission by log2-transformation.

[c] Standardized covariate of the ratio of colloid-to-crystalloid infusion by square root

[d] Internal validation was performed in training (test) sets with bootstrap (n = 500)

model showed that a high cumulative amount of colloid infusion was significantly associated with eGFR reduction in DSS patients. However, the colloid-to-crystalloid infusion ratio did not show a significant correlation. Notably, profound DSS, severe bleeding, and severe transaminitis were identified as significant prognostic indicators of a decreased eGFR. Further analysis showed good normality of the residual parameter from the linear mixed model, indicating the high validity of the developed model.

## Discussion

The fatality rate of patients with dengue shock syndrome ranges from 5% to 20% [1,2]. In particular, a dramatically higher in-hospital mortality rate (> 30%) has been observed in DSS patients with prolonged shock, dengue-associated acute liver failure, and severe respiratory failure requiring mechanical ventilation [3–5]. Although colloidal solutions are recommended

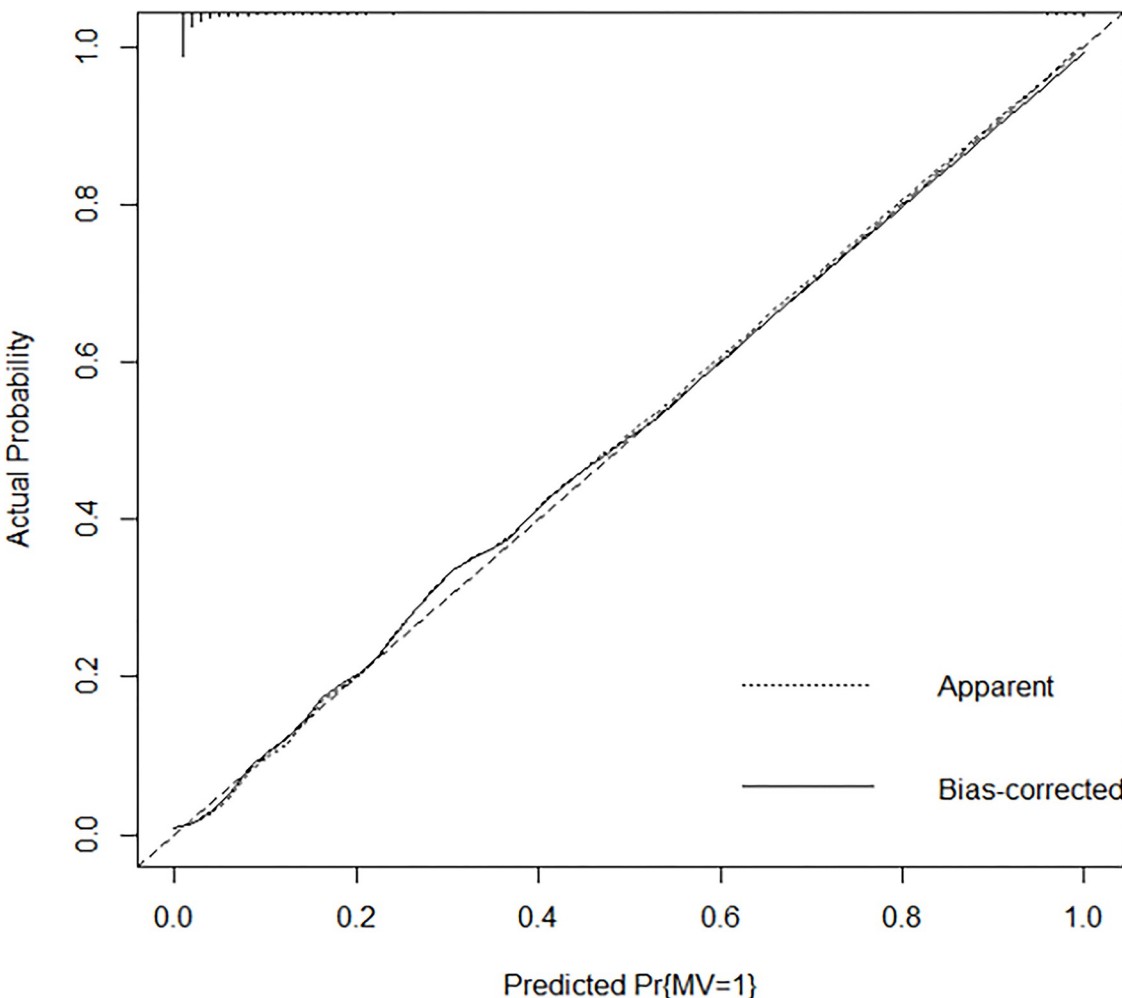

**Fig 3. Calibration plot for the predictive model of mechanical ventilation in patients with dengue shock syndrome.** The developed prognostic model showed good consistency between the predicted values (x-axis) and observed data (y-axis).

for treating patients with severe and prolonged DSS [8], an excessive amount of intravenous (IV) fluid infusion, particularly colloidal solutions, is strongly associated with adverse outcomes among patients with DSS [3,6,9,10]. Therefore, we investigated the associations and prognostic values of resuscitation fluid load and the percentages of colloid and crystalloid solutions, particularly the colloid-to-crystalloid infusion ratio, in relation to clinical outcomes, such as MV requirement and reduced renal function, in children with DSS admitted to the PICU.

In this study, 13.3% of the patients with DSS developed severe respiratory failure requiring MV within 48 h of PICU admission. The overall in-hospital mortality rate was 3.1%. Critically, the fatality rate among patients with severe and prolonged DSS who required MV was as high as 22.4%. This cohort exhibited salient features, including profound DSS with progressively massive plasma leakage and elevated intra-abdominal and thoracic pressures [2,4]. Such conditions lead to a significant preload reduction, necessitating large volumes of resuscitation fluids, ultimately contributing to multiorgan damage and death [4,20]. Notably, a meta-analysis has shown that resuscitation using macromolecular solutions is associated with increased mortality and acute kidney injury (AKI); nevertheless, better clinical outcomes have been reported in patients treated with hyperoncotic albumin [21].

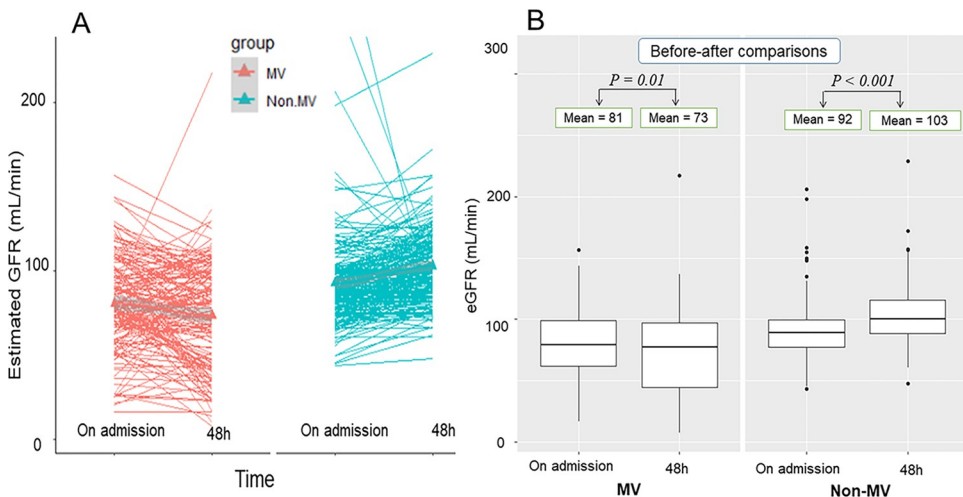

**Fig 4. Dynamic changes in estimated glomerular filtration rates between the MV and non-MV groups at PICU admission and 48 h post-admission.**

Our study highlighted notable findings considering that the patient cohort experienced prolonged DSS and required MV. This group was resuscitated with large volumes of colloid solutions, resulting in a higher colloid-to-crystalloid ratio than that in the non-MV group. Our study shows that total cumulative fluid infusion within the first 24 h of PICU admission is a strong predictor of mechanical ventilation, which is consistent with previous reports [6,22,23]. Importantly, administering macromolecular solutions to severe DSS patients with extensive plasma leakage, who do not respond to initial crystalloid resuscitation, is often inevitable.

**Table 4. Linear mixed-effects model of for changes in glomerular filtration rate on PICU admission and 48h post-admission in patients with DSS (n = 377).**

| Random effect analysis | Variance | Standard deviation | | |
|---|---|---|---|---|
| Groups (MV and Non-MV) | | | | |
| Intercept | 39 | 6.2 | | |
| Residuals [a] | 721 | 27 | | |
| **Fixed effect analysis** | **Estimate** | **SE** | **95% CI** | **P-value** |
| Intercept | 90.3 | 5.0 | 80.5 to 100.1 | 0.01 |
| Square root of cumulative amount of colloid fluid infused during 24h admission (mL/kg) [b] | 0.84 | 0.39 | 0.08 to 1.60 | 0.03 |
| Square root of ratio of colloid-to-crystalloid fluid infusion [b] | - 2.64 | 1.97 | -6.5 to 1.22 | 0.18 |
| Profound DSS | - 15.24 | 3.44 | -22 to -8.5 | < 0.001 |
| Severe bleeding | - 7.94 | 3.10 | -14 to -1.86 | 0.01 |
| Severe transaminitis | - 6.86 | 3.02 | -12.8 to 0.94 | 0.02 |

**Abbreviations:** CI, 95% confidence interval; DSS, Dengue shock syndrome; MV, mechanical ventilation; PICU, Pediatric intensive care unit; SE, Standard error

[a] Random effect was adjusted for mechanical ventilation

[b] Standardisation of covariates by the square root transformation

However, a high volume of colloid solution may pose a significant risk of severe respiratory failure, necessitating mechanical ventilation. Specifically, our study found that a colloid-to-crystalloid ratio cut-off point of $\geq 1.6$ yielded a high predictive value with an AUC of 78.1% for MV requirement. These results highlight several important findings. First, the large volumes of macromolecular solutions administered to patients with severe DSS for many consecutive hours were due to progressively extensive plasma leakage. Second, in patients with prolonged and recurrent dengue shock, early fluid resuscitation with a large colloid volume significantly correlated with mechanical ventilation support. Lastly, inappropriate and prolonged use of colloid solutions beyond the actual demand increases the risk of MV in DSS patients.

Based on our findings and published reports [2], we assumed that increasing hematocit might be a protective factor against the need for MV, and restricting IV fluids has been proposed for mechanically ventilated DSS patients. These assumptions should be investigated in future studies. In addition, the use of colloidal solutions should be minimized to ensure adequate hemoperfusion. Furthermore, a portion of the colloid fluid should be replaced early with hyperoncotic albumin or fresh frozen plasma in cases of severe bleeding and coagulation disorders. A retrospective study by Lalitha *et al.* on 76 children with DSS showed that 20% albumin rapidly decreased lactate levels and shortened the duration of mechanical ventilation and hospital stay compared with 5% albumin [24]. Notably, albumin potentially protects and maintains the integrity of the endothelial glycocalyx layer, reducing the rate of plasma leakage into interstitial tissues and restoring microcirculatory hemoperfusion [25]. In particular, concentrated albumin helps reabsorb fluid from the interstitial tissue, restore circulatory volume, and reduce fluid demand [26,27].

To the best of our knowledge, this is the first study to examine the colloid-to-crystalloid ratio to predict the risk of mechanical ventilation in pediatric patients with DSS. When fluid resuscitation with colloidal solutions is mandatory for patients with severely prolonged DSS, restricting the amount of colloidal fluid can be challenging. This is particularly true when patients experiencing hypotensive shock receive high doses of colloids soon after the initial failure of crystalloid fluid resuscitation [7]. The colloid-to-crystalloid ratio serves as a significant indicator for guiding colloid fluid resuscitation, and future trials should explore the optimal colloid-to-crystalloid ratio and the role of other fluids, such as hyperoncotic albumin. Additionally, colloid solutions potentially cause coagulation disorders and pose a higher risk of bleeding, which may necessitate additional blood product transfusions [28]. Excessive hemodilution from large amounts of colloid solutions can paradoxically decrease oxygen delivery (DO2) by reducing hemoglobin levels, complicating the identification of severe bleeding and potentially leading to unnecessary transfusions [29]. This can increase the risk of mechanical ventilation and mortality owing to fluid overload.

Importantly, it has been reported that approximately 4% of hospitalized patients with severe dengue infection develop acute kidney injury (AKI), with 14.1% requiring continuous renal replacement therapy for critical AKI [12]. In this study, 164 of 1,278 participants (12.8%) experienced severe, prolonged, and recurrent dengue shock, necessitating high doses of colloid fluid resuscitation. However, excessive use of colloidal fluids may increase the risk of AKI and coagulation disorders [30,31]. Clinical data on AKI due to colloid overuse in patients with severe DSS undergoing MV remain limited. Therefore, we aimed to explore the impact of high colloid fluid infusion on renal function within 48 hours of PICU admission in patients with profound DSS with and without MV support. Notably, our study also included severe DSS patients who were referred from provincial hospitals and had already received large volumes of IV fluid complicated by severe respiratory failure. The most important finding of this study is that children with DSS who experienced severe respiratory failure requiring mechanical

ventilation were more likely to have acute kidney injury, as indicated by decreased eGFR, than those who did not require MV support. Conversely, DSS patients without MV support responded well to fluid resuscitation and recovered quickly, as evidenced by the rapid improvement in kidney function 48 h post-PICU admission. One plausible explanation is that, compared to DSS patients without MV support, those with MV received a substantially higher volume of resuscitation fluids, including a higher percentage of colloids. Another important point is that the random effects analysis revealed that only a 5% difference in eGFR was explained by mechanical ventilation. This suggests that other determinants from fixed-effect modeling, including severe bleeding, severe transaminitis, and severe prolonged DSS, have played a more prominent role. These factors are crucial in the pathogenesis of severe dengue and serve as significant predictors of MV in children with DSS [2,4,6]. Our findings on the prognostic factors for AKI in severe dengue patients are consistent with those of previous reports [11–14]. Possible causes of AKI in children with DSS include (i) pre-renal factors such as intravascular volume inadequacy, hypovolemia, and reduced hemoperfusion to the kidneys due to severe bleeding and progressive vascular leakage in severe DSS patients, and (ii) at-renal indicators such as proximal tubule injury and damage caused by the high load of colloid solutions administered to patients with severe and prolonged DSS [9,10]. Both Wang *et al.* and Diptyanusa *et al* [12,14] reported significant case fatality rates in DSS patients with AKI requiring dialysis, ranging from approximately 14% to 22%. In contrast, most patients in our study with severe DSS and AKI survived until hospital discharge. The major causes of DSS in our study were acute respiratory distress syndrome (ARDS), pulmonary and cerebral hemorrhage, dengue-associated acute liver failure, and multiorgan failure [4,5]. Notably, our study had distinct characteristics of participants compared to previous studies conducted by Wang *et al.* and Diptyanusa *et al.* [12,14]. First, our study cohort included pediatric patients (aged < 18 years) with DSS compared to adult dengue cohorts in those studies [12,14]. Second, in the study by Wang *et al.* [14], a higher proportion of underlying diseases, including hypertension, nephrotoxic drug use, and hematuria, were independent risk factors for AKI. However, these factors were not observed in our pediatric cohort. In addition, our study revealed that profound DSS, critical bleeding, and severe transaminitis are significant risk factors for AKI events, which is consistent with the report by Diptyanusa *et al.* [12]. Lastly, compared to previous studies, our study had the strength of describing the dynamic variation of kidney function (eGFR); however, the data were only available for 373 (of the total 1,278 patients). Hence, this study limitation impeded the accurate calculation and reporting of AKI prevalence in our patient cohort. Remarkably, our research group recently reported that the cumulative amount of infused fluid is a significant predictor of in-hospital mortality in children with DSS [15]. In this study, we further identified that a large cumulative amount of intravenous fluid infusion ($\geq$ 181 mL/kg), particularly with a high proportion of colloid solutions, was associated with adverse dengue-related complications including mechanical ventilation requirement and acute kidney injury. Notably, higher fatality rates have been reported in patients with dengue who experience acute respiratory failure and AKI [3,13]. In clinical practice, a small proportion of patients with severe and prolonged DSS experience progressively massive plasma leakage, leading to a substantial increase in thoracic pressure and the need for MV [4,20]. In such cases, an intensive fluid intervention scheme with hemodynamic monitoring, preferably under point-of-care ultrasound (POCUS) guidance, is recommended. A higher volume of resuscitation fluid has been shown to yield better survival outcomes based on improvements in serum lactate levels, vasoactive inotropic score, and organ dysfunction [4]. However, this approach should be cautiously considered, as most patients in that study cohort were in the very late stage of recurrent and/or profound DSS with extensive vascular leakage and preload insufficiency [4]. For the general population with dengue shock syndrome, minimal fluid

administration to ensure tissue perfusion remains the cornerstone of dengue treatment, in line with the WHO 2009 dengue guidelines [7]. Additional resuscitation fluid should only be administered to confer a survival benefit in a small subset of patients experiencing dengue-associated obstructive shock syndrome [4,20]. Further comprehensive studies focusing on this critical population are needed to provide more convincing evidence.

Our study had several limitations inherent to its single-center, retrospective cohort design and non-standardized collection of clinical and laboratory data during hospital admission. A significant potential source of information bias arises from differences in disease severity assessments and fluid management between patients referred from provincial hospitals (who had already received resuscitation fluids) and those directly admitted to our hospital. The referred group commonly presented with more severe dengue, such as prolonged and severe DSS, than the direct admission group. Despite these limitations, our study population was relatively representative and generalizable. Further investigations in prospective cohorts with appropriate study designs are essential to validate our findings.

This study has shown the significance of optimizing fluid management protocols among hospitalized children with severe DSS in improving clinical outcomes, including MV and acute kidney injury. Our study also raised several important questions: (i) permissively elevated hematocrit levels and clinical outcomes in patients with severe dengue, (ii) the beneficial role of hyperoncotic albumin in patients with severe DSS, and (iii) the optimal volumes of colloid and albumin solutions administered to patients with severe DSS. Further investigation in clinical trials is essential to elucidate this knowledge gap, which is required to optimize the protocol for fluid resuscitation among patients with severe DSS.

## Conclusion

In conclusion, a significant volume of resuscitation fluid ($\geq$ 181 mL/kg) and a higher colloid-to-crystalloid infusion ratio ($\geq$ 1.6) were associated with adverse outcomes, including the need for mechanical ventilation and acute kidney injury, within the first 48 hours of admission to the PICU. Other critical prognostic factors included profound dengue shock, severe transaminitis, and critical bleeding. These findings are essential for optimizing fluid management protocols to enhance the clinical outcomes in patients with dengue shock syndrome.

## Supporting information

**S1 Checklist. STROBE checklist.**
(DOC)

**S1 File. Indications for mechanical ventilation support in children with severe DSS.**
(DOCX)

**S1 Appendix. Observational data of 1,278 participants in the study cohort.**
(XLSX)

## Acknowledgments

We are grateful to the patients and administrative staffs, particularly Drs DTT Hien and TTTri for their support with this study.

## Author Contributions

**Conceptualization:** Vo Thanh Luan, Trinh Huu Tung, Nguyen Tat Thanh.

**Data curation:** Vo Thanh Luan, Vo Thi-Hong Tien, Ngo Thi-Mai Phuong, Nguyen Tat Thanh.

**Formal analysis:** Nguyen Tat Thanh.

**Funding acquisition:** Vo Thanh Luan, Do Chau Viet.

**Investigation:** Vo Thanh Luan, Vo Thi-Hong Tien, Ngo Thi-Mai Phuong, Do Chau Viet, Nguyen Tat Thanh.

**Methodology:** Vo Thanh Luan, Nguyen Tat Thanh.

**Project administration:** Vo Thanh Luan, Do Chau Viet, Trinh Huu Tung.

**Resources:** Vo Thanh Luan, Do Chau Viet, Trinh Huu Tung.

**Supervision:** Vo Thanh Luan, Do Chau Viet, Trinh Huu Tung.

**Writing – original draft:** Vo Thanh Luan, Nguyen Tat Thanh.

**Writing – review & editing:** Vo Thanh Luan, Do Chau Viet, Trinh Huu Tung, Nguyen Tat Thanh.

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
