## [Decision Letter · Decision Letter 0]

22 Oct 2024

Dear Mr Nguyen,

Thank you very much for submitting your manuscript "Associations of resuscitation fluid load, colloid-to-crystalloid infusion ratio and clinical outcomes in children with dengue shock syndrome" for consideration at PLOS Neglected Tropical Diseases. As with all papers reviewed by the journal, your manuscript was reviewed by members of the editorial board and by several independent reviewers. In light of the reviews (below this email), we would like to invite the resubmission of a significantly-revised version that takes into account the reviewers' comments. 

We cannot make any decision about publication until we have seen the revised manuscript and your response to the reviewers' comments. Your revised manuscript is also likely to be sent to reviewers for further evaluation.

Sincerely,

Feng Xue, Ph.D.

Guest Editor

Elvina Viennet

Section Editor

Reviewer's Responses to Questions

**Key Review Criteria Required for Acceptance?**

**Methods**

-Are the objectives of the study clearly articulated with a clear testable hypothesis stated?

-Is the study design appropriate to address the stated objectives?

-Is the population clearly described and appropriate for the hypothesis being tested?

-Is the sample size sufficient to ensure adequate power to address the hypothesis being tested?

-Were correct statistical analysis used to support conclusions?

-Are there concerns about ethical or regulatory requirements being met?

Reviewer #1: Yes

Yes

Yes; Yes

Yes

Yes

No

Reviewer #2: The objectives are clearly stated, study design is adequate, study population is well described and statistical analyses seem appropriate.

Reviewer #3: (No Response)

Reviewer #4: Objectives are stated clearly but hypothesis is not stated.

Appropriate study design 

Population is clear and appropriate

Sample size is sufficient

Statistical analysis is adequate

No ethical concerns

Reviewer #5: (No Response)

**Results**

-Does the analysis presented match the analysis plan?

-Are the results clearly and completely presented?

-Are the figures (Tables, Images) of sufficient quality for clarity?

Reviewer #1: yes

yes

yes

Reviewer #2: Results are well presented, figures and tables are appropriate.

Reviewer #3: (No Response)

Reviewer #4: Strengths:

Detailed Data: The table captures a wide range of clinical and laboratory data, providing a comprehensive snapshot of the study population.

Clear Metrics: The use of median and interquartile ranges (IQR) for continuous variables ensures a clear understanding of data distribution.

Relevant Variables: Including key variables such as age, sex, DSS severity, and various clinical measures gives a thorough overview of the patients’ condition upon PICU admission.

Areas for Improvement:

Elaborate on Significance: It would be useful to include a brief explanation of why certain variables (like the systolic and diastolic shock indices or the high cumulative IV fluids) are significant and how they relate to patient outcomes.

Comparative Context: Providing comparative data or referencing other studies could help put these findings into context. How do these figures compare with similar studies or expected norms?

Visual Aids: Including charts or graphs alongside the table could help visualize key trends and differences, making the data more digestible.

Suggestions:

Consider discussing the implications of these baseline characteristics for the prognosis or treatment strategies of DSS patients.

Discuss any potential biases introduced by the selection of participants and how these were managed or accounted for in the analysis.

Reviewer #5: (No Response)

**Conclusions**

-Are the conclusions supported by the data presented?

-Are the limitations of analysis clearly described?

-Do the authors discuss how these data can be helpful to advance our understanding of the topic under study?

-Is public health relevance addressed?

Reviewer #1: yes

yes

yes

no

Reviewer #2: In the discussion section, the authors make several claims that cannot be directly inferred from the results. It is crucial to review certain sentences to align the conclusions with what was actually studied. While the formulated hypotheses are entirely plausible, it should be explicitly stated that they are only hypotheses addressing a complex phenomenon involving simultaneous inflammatory and hemodynamic changes, all contributing to the primary outcome: the requirement for mechanical ventilation. I would recommend more caution when interpreting therapeutic findings from observational studies, due to the numerous confounders present. Predictors of severity should not be mistaken for causative factors.

Early in the discussion, the authors state that "Resuscitation using macromolecular solutions is associated with increased mortality and acute kidney injury (AKI), whereas improved outcomes have been reported in patients treated with hyperoncotic albumin [21]" (line 326). However, since the use of albumin is mentioned only in Table 2 as one of the blood products and not discussed earlier in the paper, emphasizing this hypothesis as an explanation for the results is unexpected.

In discussing the results (line 332), the authors appropriately note that their study demonstrates total cumulative fluid infusion within the first 24 hours of PICU admission is a strong predictor of mechanical ventilation. They point out that administering macromolecular solutions to severe DSS patients with extensive plasma leakage, who do not respond to initial crystalloid resuscitation, is often inevitable. Later, they state that “the high volume of colloid solution poses a significant risk of severe respiratory failure, necessitating mechanical ventilation” (line 336). However, it is not possible to assert that the volume of colloid is a cause of respiratory failure, or whether it is merely a predictor MV. The high volume of colloid solution may simply indicate severe disease, characterized by a high inflammatory response and significant plasma leakage.

Similarly, the finding that "rising hematocrit was a protective factor against the need for MV" (line 345) is insufficient to support the statement that "This indicates that restricting IV fluids is recommended, despite the relatively elevated hematocrit levels.", as it is not possible to assert that the elevation in hematocrit is a consequence of fluid restriction or merely a sign of clinical recovery, with improved renal function and hemodynamics, leading to a decreased need for fluid infusion. The same applies to the recommendation in line 347 that colloidal solutions should be minimized to ensure adequate hemoperfusion. This hypothesis could be tested in future studies, but it is not a direct conclusion of the current study, as it was not evaluated. 

In line 362, instead of saying, "The colloid-to-crystalloid ratio serves as a significant indicator for guiding colloid fluid resuscitation and suggests the optimal time to substitute colloid solutions with other beneficial fluids, such as hyperoncotic albumin," it would be more accurate to say that the colloid-to-crystalloid ratio serves as a significant indicator for guiding colloid fluid resuscitation, and that future trials should explore the optimal colloid-to-crystalloid ratio and the role of other fluids, such as hyperoncotic albumin.

Similarly, it is not possible to conclude that the need for more blood transfusions among patients requiring MV support is due to the excessive use of macromolecular solutions, as suggested by the phrase on line 370. More likely, these needs arise from a common cause explained by the pathophysiology of profound DSS. Furthermore, as albumin is among these blood products, it complicates the evaluation of this finding.

Reviewer #3: (No Response)

Reviewer #4: Strengths:

Clarity: You’ve succinctly highlighted the key findings and their clinical relevance.

Practical Application: Emphasizing the need for optimizing fluid management protocols demonstrates the importance of your research.

Areas for Improvement:

Specificity: Consider providing more specific recommendations based on your findings. For instance, detailing how fluid management protocols should be adjusted.

Future Research: Briefly mentioning areas for future research could strengthen the conclusion. This shows awareness of the study's limitations and an understanding of the broader research landscape.

Reviewer #5: (No Response)

**Editorial and Data Presentation Modifications?**

Reviewer #1: I have added an attachment to the article with suggested edits for the author

Some errors in the article as outlined. Overall, The study is important and adds to the literature available for treatment or lack there of on the topic of DSS.

Limited impact and generalizability of the study results

Reviewer #2: I would recommend Minor Revision. 

The results are relevant and methods seem adequate. 

Although results are not sufficient to propose new treatment guidelines, they indicate prognostic predictors and can support future clinical trials.

Reviewer #3: (No Response)

Reviewer #4: Minor Revision

Reviewer #5: (No Response)

**Summary and General Comments**

Reviewer #1: The study is important and adds to the literature available for treatment or lack there of on the topic of DSS but the disease is endemic to Asian countries, mostly South East Asia which limits the impact and generalizability of the study results.

The data is repetitive and a lot of the same things are mentioned in various different places in the study.

Reviewer #2: The paper addresses an important issue, analyzing a large case series of children with DSS. It clearly outlines the methods and acknowledges its limitations. The results are well-presented. However, the discussion section requires some revision. The findings related to prognostic predictors of fluid resuscitation parameters are crucial for clinical management and can inform future clinical trials, though they cannot be directly translated into treatment recommendations.

Reviewer #3: The topic addresses a critical and specific area of pediatric care for DSS patients, adding valuable insights to the body of literature on dengue-related complications. This study is among the first to examine the colloid-to-crystalloid ratio as a predictor of mechanical ventilation in DSS, representing a novel and important contribution. It provides clear statistics, such as the fatality rate among patients requiring MV (22.4%) and the predictive value of the colloid-to-crystalloid ratio for MV requirement. The manuscript discusses the clinical implications of fluid resuscitation in DSS, particularly highlighting the risks of using large volumes of colloid solutions and the potential for acute kidney injury (AKI) and mechanical ventilation—crucial information for clinical practice. However, several points require further attention:

1. Some sentences are overly complex and could be simplified for better readability. For instance:

•The sentence "Resuscitation using macromolecular solutions is associated with increased mortality and acute kidney injury (AKI), whereas improved outcomes have been reported in patients treated with hyper-oncotic albumin" can be rewritten for clarity.

•Additionally, consider breaking long paragraphs into smaller ones, especially when transitioning between key ideas. For example, the paragraph starting with "In this study, 13.3% of the patients with DSS developed severe respiratory failure..." should be divided when discussing different risk factors for mortality and mechanical ventilation.

2. The manuscript would benefit from more detailed descriptions of the cohort, particularly in terms of demographics and baseline health conditions. This additional context would help readers assess whether the findings can be generalized to broader populations. This is especially important given the mention of the patient population being derived from both provincial hospitals and direct admissions, which suggests heterogeneity in disease severity.

3. While the limitations section is included, it should be expanded. The retrospective design is noted, but other potential confounders—such as variations in treatment protocols between different hospitals—should be discussed more thoroughly. The manuscript would also benefit from addressing selection bias due to the referral of more severe cases to the PICU. Moreover, the differences between patients referred from provincial hospitals and those admitted directly should be explored in more depth to assess how this variability might influence the study's findings.

4. Although the discussion references other studies, such as those by Wang et al. and Diptyanusa et al., a clearer comparison between this study's findings and previous research would strengthen the manuscript. How do your results regarding AKI and MV align or contrast with existing studies? Providing this context could enhance the study's contribution to the field.

5. The colloid-to-crystalloid ratio as a predictor of MV is a key point, but the discussion would benefit from a more detailed analysis of why a ratio of ≥1.6 is significant. How does this threshold compare with other fluid resuscitation guidelines? A more explicit discussion of potential cutoff points and alternative resuscitation strategies is needed.

6. The clinical recommendations regarding the minimization of colloid use and the potential role of hyperoncotic albumin are important but could be more specific. For instance, in which clinical scenarios should albumin be preferred over other fluids? Are there any guidelines or algorithms that clinicians should follow based on the study’s findings? Providing these details would make the recommendations more actionable.

Reviewer #4: Overall Comments:

Comprehensive Coverage: Your paper does an excellent job of covering all necessary sections from the methodology to the results and conclusions. Each part is meticulously detailed, which is crucial for understanding the context and findings of your study.

Ethical Considerations: The ethical approval and adherence to the guidelines are well-documented. This strengthens the credibility of your research.

Data Analysis: Your statistical analysis is robust and well-justified. The use of various transformation and validation methods adds depth to your analysis, ensuring the reliability of your findings.

Clinical Relevance: The conclusions drawn are directly applicable to clinical practice, which is a significant strength. Your emphasis on optimizing fluid management protocols is particularly impactful.

Clear Presentation: Overall, your presentation of data, including tables and figures, is clear and easy to follow. However, integrating visual aids such as flowcharts or graphs to complement the data tables could enhance understanding.

Suggestions for Improvement:

Contextual Background: Providing a bit more context in the introduction about the significance of studying DSS in the population can help readers unfamiliar with the region or the condition.

Limitations: While you’ve mentioned missing data and the retrospective nature of the study, a more detailed discussion on the limitations and potential biases could be beneficial.

Future Research: Identifying areas for future research based on your findings can add value, highlighting how your study paves the way for further investigation.

Your research is well-structured and addresses important clinical questions. A few tweaks here and there, and you’re all set for a great impact! Keep up the good work!

Reviewer #5: COMMENTS 

This retrospective study was conducted at a tertiary pediatric hospital in Vietnam from 2013 to 2022. It explores the associations and prognostic significance of resuscitation fluid load and, the colloid-to-crystalloid infusion ratio, in relation to clinical outcomes such as the need for mechanical ventilation and impaired renal function in children with dengue shock syndrome admitted to the PICU. The study provides valuable insights that can inform future randomized controlled trials in this vulnerable population. 

Specific comments: 

Introduction:

- Discuss the ongoing debate regarding the ideal type and volume of fluids, as well as the optimal balance between colloid and crystalloid administration.

- Conclude the introduction with a compelling statement highlighting the significance of the study. Emphasize why this research is essential and how it can help overcome the challenges of managing DSS in the PICU.

Methods and results: 

- Since this is retrospective data, please mention the data source (electronic or paper-based), its reliability, and how the data was retrieved.

- Excluding participants with missing data can introduce selection bias, as those included in the analysis may differ systematically from those excluded.

- When describing the models, provide a brief explanation for the inclusion of specific variables. For instance, clarify why variables like severe bleeding, severe transaminitis, hematocrit, and platelet counts were chosen. This helps readers understand the rationale behind your model choices. The severity of DSS, as well as the group referred from other hospitals, could be potential confounders that might influence the need for mechanical ventilation or the risk of kidney injury.

- Please clarify how the authors determined the sample size required to detect statistically significant effects between resuscitation fluid load, the colloid-to-crystalloid infusion ratio, and clinical outcomes in children with DSS. Alternatively, conduct a power analysis to assess whether the dataset is sufficient to test the hypotheses adequately.

- The authors mentioned that no significant interactions were found among the covariates, but no data was provided to substantiate this claim.

- In Table 4, the authors reported that the random effects analysis revealed an intra-group correlation of 5%, but no supporting documentation was provided. The authors should include the full model as a supplementary table so that interested readers can examine the results in detail.

Discussion

- Avoid referencing tables in the discussion section.

- The authors identified female gender as a significant predictor of mechanical ventilation in DSS patients, which requires further explanation.

- Discuss how these data can be helpful to advance our understanding of fluid management in DSS.

- Discuss the public health relevance of the study results?

- Consider discussing future research directions, for example, the types of colloid and crystalloid fluids to be used for resuscitation and the optimal timing for administering each.

PLOS authors have the option to publish the peer review history of their article (what does this mean?). If published, this will include your full peer review and any attached files.

Reviewer #1: Yes: Sanwal Singh Mehta

Reviewer #2: No

Reviewer #3: No

Reviewer #4: Yes: Eloho Patricia Hambolu

Reviewer #5: No
---

## [Decision Letter · Decision Letter 1]

16 Dec 2024

Dear Mr Thanh ,

We are pleased to inform you that your manuscript 'Associations of resuscitation fluid load, colloid-to-crystalloid infusion ratio and clinical outcomes in children with dengue shock syndrome' has been provisionally accepted for publication in PLOS Neglected Tropical Diseases.

Best regards,

Feng Xue, Ph.D.

Guest Editor

Elvina Viennet

Section Editor

Shaden Kamhawi

co-Editor-in-Chief

Paul Brindley

co-Editor-in-Chief

Reviewer's Responses to Questions

**Key Review Criteria Required for Acceptance?**

**Methods**

-Are the objectives of the study clearly articulated with a clear testable hypothesis stated?

-Is the study design appropriate to address the stated objectives?

-Is the population clearly described and appropriate for the hypothesis being tested?

-Is the sample size sufficient to ensure adequate power to address the hypothesis being tested?

-Were correct statistical analysis used to support conclusions?

-Are there concerns about ethical or regulatory requirements being met?

Reviewer #2: (No Response)

Reviewer #3: All corrections are done and I have no further questions

Reviewer #4: (No Response)

Reviewer #5: The study objectives and design are appropriate for the study.

**Results**

-Does the analysis presented match the analysis plan?

-Are the results clearly and completely presented?

-Are the figures (Tables, Images) of sufficient quality for clarity?

Reviewer #2: (No Response)

Reviewer #3: All corrections are done and I have no further questions

Reviewer #4: (No Response)

Reviewer #5: Yes

**Conclusions**

-Are the conclusions supported by the data presented?

-Are the limitations of analysis clearly described?

-Do the authors discuss how these data can be helpful to advance our understanding of the topic under study?

-Is public health relevance addressed?

Reviewer #2: At the end of Discussion session, I would suggest to replace "prospective cohorts" by "clinical trials" in the following sentence:

"Further investigation in prospective cohorts is essential to elucidate this knowledge gap, which is required to optimize the protocol of fluid resuscitation among patients with severe DSS."

Although observational studies can offer valuable insights, they are more susceptible to bias and confounding variables, which makes it difficult to establish a clear cause-and-effect relationship. Therefore, clinical trials are the gold standard for assessing the effectiveness of treatment protocols.

Reviewer #3: All corrections are done and I have no further questions

Reviewer #4: (No Response)

Reviewer #5: Yes

**Editorial and Data Presentation Modifications?**

Reviewer #2: (No Response)

Reviewer #3: (No Response)

Reviewer #4: (No Response)

Reviewer #5: (No Response)

**Summary and General Comments**

Reviewer #2: The authors have met all recommendations regarding critical points

Reviewer #3: (No Response)

Reviewer #4: (No Response)

Reviewer #5: Authors tried to address all comments.

PLOS authors have the option to publish the peer review history of their article (what does this mean?). If published, this will include your full peer review and any attached files.

Reviewer #2: **Yes: **Regina Paiva Daumas

Reviewer #3: No

Reviewer #4: **Yes: **Eloho Hambolu

Reviewer #5: No

---

## [Editor Report · Acceptance letter]

30 Dec 2024

Dear Mr Thanh ,

We are delighted to inform you that your manuscript, "Associations of resuscitation fluid load, colloid-to-crystalloid infusion ratio and clinical outcomes in children with dengue shock syndrome," has been formally accepted for publication in PLOS Neglected Tropical Diseases.

Best regards,

Shaden Kamhawi

co-Editor-in-Chief

Paul Brindley

co-Editor-in-Chief
